## METHOD

# A large-scale genomically predicted protein mass database enables rapid and broad-spectrum identification of bacterial and archaeal isolates by mass spectrometry

Yuji Sekiguchi[1][*] , Kanae Teramoto[2], Dieter M. Tourlousse[1], Akiko Ohashi[1], Mayu Hamajima[1], Daisuke Miura[1], Yoshihiro Yamada[3], Shinichi Iwamoto[3] and Koichi Tanaka[3]

*Correspondence:
y.sekiguchi@aist.go.jp

[1] Biomedical Research Institute, National Institute of Advanced Industrial Science and Technology (AIST), AIST Tsukuba Central 6, Ibaraki 305-8566, Japan
[2] MS Business Unit, Shimadzu Corporation, Kyoto, Japan
[3] Koichi Tanaka Mass Spectrometry Research Laboratory, Shimadzu Corporation, Kyoto, Japan

## Abstract

MALDI-TOF MS-based microbial identification relies on reference spectral libraries, which limits the screening of diverse isolates, including uncultured lineages. We present a new strategy for broad-spectrum identification of bacterial and archaeal isolates by MALDI-TOF MS using a large-scale database of protein masses predicted from nearly 200,000 publicly available genomes. We verify the ability of the database to identify microorganisms at the species level and below, achieving correct identification for > 90% of measured spectra. We further demonstrate its utility by identifying uncultured strains from mouse feces with metagenomics, allowing the identification of new strains by customizing the database with metagenome-assembled genomes.

**Keywords:** MALDI-TOF MS, Bacterial genomes, Archaeal genomes, Microbial identification, Protein mass database, Uncultured microbes, Culturomics

## Background

Owing to advancements in DNA sequencing technologies and bioinformatics, significant progress has been made in understanding the microbial world, including uncultured microbes [1, 2]. Metagenomic studies have cataloged the vast diversity of *Bacteria* and *Archaea* and revealed that most microorganisms in most ecosystems remain uncultured [3–5]. Within the Genome Taxonomy Database (GTDB), roughly half of bacterial and archaeal taxa identified at the species level are represented solely by uncultured microorganisms [6]. Even in highly studied ecosystems such as the human gut, most species remain uncultured, with more than 70% of species-level taxa in the Unified Human Gastrointestinal Genome (UHGG) collection lacking cultured representatives [4].

This has led to renewed efforts to cultivate uncultured microorganisms to directly study their functions and support their biotechnological application in areas such as

medicine [7, 8]. By using comprehensive cultivation schemes such as culturomics, hundreds to thousands of isolates are routinely generated [9]. As a result, the screening of such large libraries of isolates has become a major bottleneck. Traditionally, sequencing of the 16S rRNA gene has been the preferred approach for microbial identification, but this is difficult to apply with sufficient throughput. In comparison, matrix-assisted laser desorption/ionization time-of-flight mass spectrometry (MALDI-TOF MS) allows direct identification from whole cells with minimal sample preparation and thus offers a suitable alternative [10–13]. While already well established in clinical microbial diagnostics, the use of MALDI-TOF MS for microbial identification in microbiome research is much less widespread. Because MALDI-TOF MS relies on reference spectral libraries of cultured isolates, it often fails when applied to species not represented in such libraries, including previously uncultured microorganisms. Therefore, new strategies are needed to improve the usability of MALDI-TOF MS for broad-spectrum identification.

Here, we present a new approach and toolkit for broad-spectrum microbial identification by MALDI-TOF MS. This is enabled by a large-scale database of protein masses predicted from nearly 200,000 publicly available bacterial and archaeal genomes. This set of in silico predicted mass peak lists replaces microbial reference spectral libraries and adds thousands of species not typically covered by such libraries. Using a diverse set of 94 isolates, we validated the use of the predicted protein mass database for the identification of microorganisms at the species level and below. We further demonstrated the utility of our toolkit through the identification of 103 cultured strains from mouse feces by matching them against protein masses predicted from metagenome-assembled genomes from the same samples.

## Results

For constructing the predicted protein mass database, we downloaded all genomes, including single-cell amplified and metagenome-assembled genomes (MAGs), available in NCBI's RefSeq and GenBank databases (release 95 [14]). After screening for potentially lower-quality genome sequences (see "Database construction" section in Methods for details), a total of 193,197 genomes from 190,160 bacteria and 3037 archaea were retained (31,790 species-level taxa). Protein sequences were predicted, and theoretical molecular masses, accounting for posttranslational cleavage of N-terminal methionine and signal peptides (see "Methods" for details), in the range of 2000 to 15,000 Da (equivalent to their expected mass-to-charge ratio, *m/z*, in MALDI-TOF MS measurements), were incorporated in the database. The database, referred to as the genomically predicted theoretical protein mass database for mass spectrometry (GPMsDB), contains ~163 million protein mass entries, with an average of 845 entries per genome (interquartile range of 542–1113). These serve as theoretical mass peak lists for matching experimentally measured peak lists by MALDI-TOF MS for microbial identification.

### Resolution of microbial identification based on theoretical genome-wide protein mass profiles

We first assessed the relationship between genome-wide (that is, including all predicted proteins without theoretical mass filtering) protein mass profiles and the evolutionary relatedness of the genomes. We matched the predicted protein mass profiles

of pairs of genomes and calculated a score (scoring scheme I, see Methods for definition). To account for the mass measurement inaccuracy of MALDI-TOF MS, errors between 10 and 300 ppm were allowed for assigning matched peaks between pairs of mass protein profiles. As shown in Fig. 1 (also see Additional file 1: Figs. S1, S2, and S3), genome-wide peak matching (PM) score, which is based on the number of matched peaks between mass protein peak profiles between two genomes (see more details in "Methods" section), was related with genomic relatedness, in terms of both average nucleotide identity (ANI, or values estimated based on Mash [15] distances) and taxonomic distance as summarized in the GTDB [2, 6]. As shown in Additional file 1: Figs. S1, S2, PM scores had a strong relationship with ANI values above 80%, as well as with taxonomic ranks at the genus to strain levels. We note that relatively wide range of PM scores seen in Fig. 1 is due to differences in the relationship between PM scores and genomic relatedness among genomes, especially for genomes with a high number of genes (Additional file 1: Fig. S2). Overall, MALDI-TOF MS mass peak profiles should allow identification at the genus rank and below with mass error tolerances on the order of 200 ppm, which is the accuracy commonly achieved by widely used instruments. For genomes that differ at higher taxonomic ranks, PM scores rapidly decreased and could thus lead to unreliable identification. This also indicates that reference databases should contain representative peak lists matching microorganisms to be identified at the genus-to-species level for reliable performance. Critically, in our approach, this can be easily achieved based on genome sequences rather than experimentally acquired MALDI-TOF MS spectra.

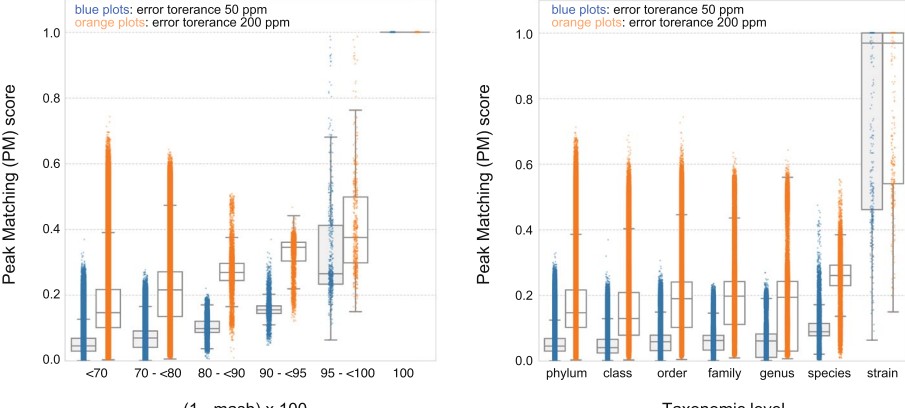

**Fig. 1** Relationship between Mash distance (**A**) or taxonomic distance (**B**) and peak matching (PM) score. For panel **A**, the inverse Mash distance ([1 − mash] × 100, nearly equivalent to ANI (%)) was calculated and binned as shown on the x-axis. For panel **B**, taxonomic distances were obtained based on the GTDB r95 taxonomy; labels on the x-axis indicate taxonomic levels at which differences in PM scores are evaluated (for example, "strain" indicates pairs of genomes within the same species). PM scores were obtained from 200 randomly selected genomes matched against all genome entries in the GPMsDB (193,197 reference genome entries). Mass error tolerances of 50 (blue) and 200 ppm (red) were used for peak matching, and the resultant values were plotted as dots for given pairs of genomes. The distribution of the quantitative data is also shown as box plots, where the quartiles of the dataset are shown as boxes, while the whiskers extend to show the rest of the distribution, except for points that were determined to be "outliers" using a method that is a function of the interquartile range

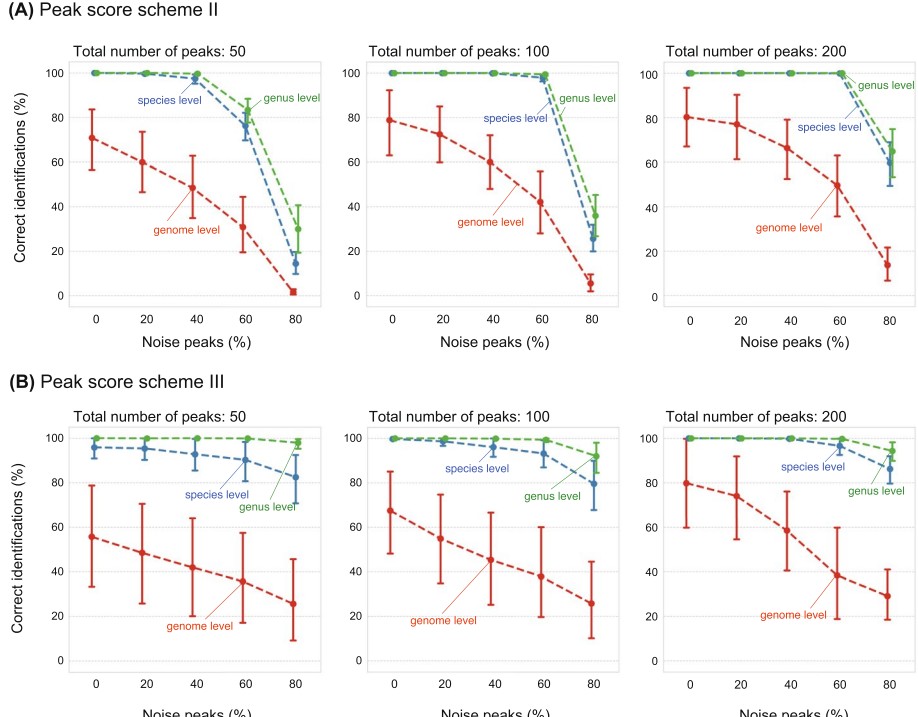

**Fig. 2** Accuracy of taxonomic identification based on PM scores as evaluated with simulated peak lists for five representative genomes (*Escherichia coli*, GCF_003697165.2; *Acinetobacter calcoaceticus*, GCF_000368965.1; *Bacillus subtilis*, GCF_000009045.1; *Cutibacterium acnes*, GCF_003030305.1; *Methanothermobacter thermautotrophicus*, GCF_000008645.1). Peak matching was performed at a mass error tolerance of 200 ppm. Simulated peak lists consist of theoretically expected mass peaks containing at least six ribosomal proteins and varying amounts of random background peaks (noise). In brief, for each genome, level of noise, and the total number of peaks, 500 peak lists were generated, and the percentage of correct identification based on the best-matching hit was calculated at different taxonomic levels (genome level: red, species level: blue, genus level: green). The data are plotted as the means of all five genomes with error bars as the standard deviations. Panels **A** and **B** show the results based on scoring schemes II and III, respectively

### Evaluation of performance using simulated peak lists

Next, we used simulated peak lists to investigate the accuracy of identification and fine-tune our peak-matching scoring scheme (Methods). For this analysis, we considered proteins with the mass range of 2000–15,000 Da. In short, we generated simulated peak lists by randomly selecting 50, 100, or 200 mass peaks from their protein mass lists. Using scoring scheme I (see above), simulated peak lists were assigned to the correct genome, species, or genus at low error tolerance (10 ppm) but failed when peak matching was performed at 200 ppm tolerance (Additional file 1: Fig. S4A). This was attributed to the sensitivity of the PM score to the number of protein masses in the reference, resulting in inflated PM scores against reference genomes with low theoretical protein counts. This could be remediated by bounding the denominator in the formula of scoring scheme I (scoring scheme II), which allowed accurate identification with tolerance up to 200 ppm (Additional file 1: Fig. S4BCD).

We next generated simulated peak lists containing additional mass peaks with random *m/z* values, mimicking noise and evaluated the identification accuracy. Even in the presence of 80% random mass peaks, good accuracy was achieved at the genus and species levels (Fig. 2 and Additional file 1: Fig. S5). Next, considering that ribosomal proteins are

typically among the most widely detected proteins in MALDI-TOF MS measurements, we further evaluated score scheme III, in which matched ribosomal proteins are given more weight. This resulted in improved accuracy of identification for simulated peak lists containing four or more peaks from ribosomal proteins (Additional file 1: Fig. S6). It should be noted that correct genome-level identification may be challenging for species with a large number of similar genome entries in the GPMnDB, such as *Escherichia coli* and *Bacillus subtilis* (Additional file 1: Figs. S5 and S6).

For identification purposes, a toolkit (GPMsDB-tk) was developed in this study based on these observations with the current GPMsDB. Our toolkit accepts mass peak lists and identifies best-matching theoretical peak lists for classification and identification at the genome level (Additional file 1: Fig. S7).

### Identification of axenic reference cultures spanning different bacterial and archaeal phyla

We validated our toolkit by applying it to pure cultures of 94 strains (84 bacteria and 10 archaea, Additional file 2: Table S1) that captured 15 different phyla within the GTDB. For strains lacking genome sequences in that database, genome sequences were newly generated and added to the database (Additional file 3: Table S2). For 13 strains of the phylum *Actinomycetota* (previously known as *Actinobacteria*, also called as *Actinomycetota* in GTDB release 95), we performed more comprehensive testing and present these results in the next section. For all other strains, samples were prepared by formic acid (FA) treatment and subjected to MALDI-TOF MS analysis, with at least four determinations for each strain. For 71 bacteria and 10 archaea, 349 (94.5%) and 350 (94.8%) out of 369 measured spectra were correctly identified at the species and genus levels, respectively, using scoring scheme III (Fig. 3; Additional file 1: Figs. S8 and S9; Additional file 4: Table S3). Even at the genome level, 72.3% of measurements resulted in correct identification. These findings confirmed that the *m/z* values of mass peaks are sufficient for accurate identification at the species or subspecies level, thus highlighting the tremendous value of genomically derived protein mass databases as the basis for microbial identification by MALDI-TOF MS.

### Identification of Actinomycetota under different culture conditions

We next sought to examine the effect of varying culturing conditions and sample pre-treatments, focusing on the abovementioned 13 actinobacterial strains as examples of Gram-positive-type cell wall bacteria, which are generally thought to be difficult to lyse (Additional file 1: Fig. S10; Additional file 5: Table S4). Overall, cultivation conditions, which included four different liquid and solid media, had only a minor impact on identification. The majority of the strains (9 out of 13 strains) grown under all the cultivation

(See figure on next page.)
**Fig. 3** MALDI-TOF MS identification of 71 bacterial strains, listed in the upper part of the heatmap. The best-matched genome entries were identified using GPMsDB-tk. Peak matching was performed at a mass error tolerance of 200 ppm, and the best-matching genomes were calculated based on scoring scheme III, specifying option -a as "all". At least four determinations were performed for each strain, and the number of best-matched hits is shown in the heatmap. Bold boxes in the heatmap represent correct identification at the species level in the GTDB r95. The best hits other than the genomes shown in the tree are grouped at the bottom as "other lineages"

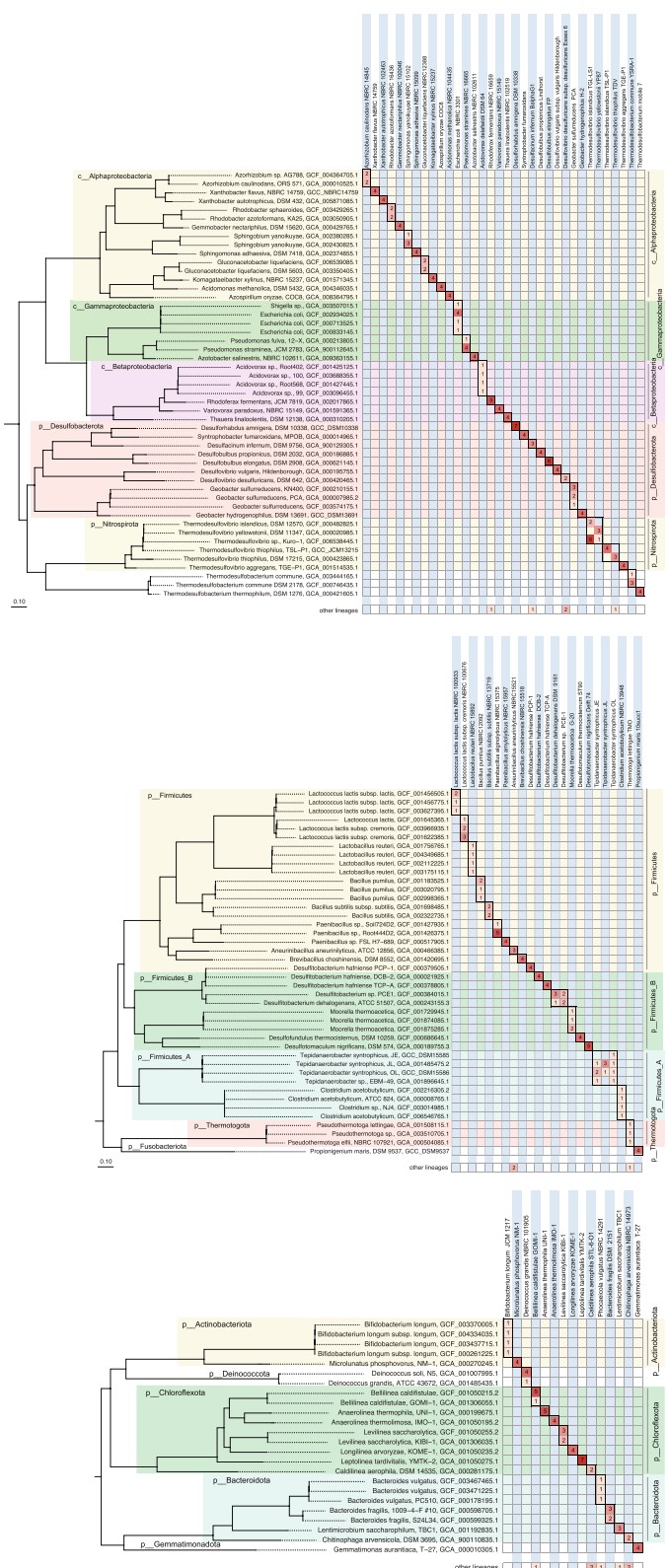

**Fig. 3** (See legend on previous page.)

conditions were correctly identified at the genus to species levels, using simple FA method as the pretreatment (see "Methods" section) (Additional file 1: Fig. S10). This confirmed that microbial identification by GPMsDB-tk is robust to differences in cultivation conditions. However, for *Glycomyces arizonensis*, *Streptomyces griseus*, *Nocardioides simplex*, and *Janibacter limosus* cells, sample treatment had a larger effect and vigorous sample pretreatment was critical for the correct identification of some strains (Additional file 1: Fig. S10; Additional file 5: Table S4). More specifically, the identification of these cells was incorrect even at the genus level when samples were prepared by simple FA treatment (Additional file 1: Fig. S10; Additional file 5: Table S4). However, in most cases, these strains could be correctly identified using more vigorous sample pretreatment involving bead-beating and heating (Additional file 1: Fig. S10; Additional file 5: Table S4). Here, we found that combined FA and heat treatment (FA heating method) resulted in the successful identification of all the actinobacterial strains tested (Additional file 1: Fig. S10). This method was further found to be suitable for the treatment of other bacterial and archaeal species, including *E. coli* (data not shown). Because of its simplicity, the FA heating method is universally applicable for a range of bacteria and archaea.

### Validation using externally obtained MALDI-TOF MS profiles

We further assessed the performance of our toolkit by applying it to two sets of MALDI-TOF MS datasets that were obtained externally, as additional examples. These captured 74 strains within the genus *Acinetobacter* (616 peak lists, SAC001), possessing > 30 different species, and 24 strains of the species *Cutibacterium acnes*, spanning three different genotypes (Additional file 1: Fig. S13). For *Acinetobacter*, 564 (91.6%) and 604 (98.1%) out of 616 available peak lists were correctly assigned at the species and genus levels, respectively (Additional file 1: Figs. S11 and S12). Importantly, this included multiple strains lacking genome sequences in the database. For *C. acnes*, 100% accurate identification was achieved at the species and genus levels. Furthermore, at the subspecies level, 110 out of 118 peak lists were correctly assigned to three genotypes (Additional file 1: Fig. S13). These data confirmed again that the toolkit can achieve accurate subspecies level identification.

### Cultivation and identification of strains from mouse fecal samples

Our toolkit enables users to construct customized databases and add new genome sequences to existing databases and can thus streamline workflows that perform shotgun metagenomic analysis and MAG reconstruction of samples of interest. We performed shotgun metagenomics on fecal samples from mice and reconstructed genomes, which yielded 84 high-quality genome bins with genome completeness of > 50% and contamination of < 10% (Additional file 6: Table S5). In the cultivation of anaerobes from the same sample, > 100 colonies were selected and anaerobically subcultured in 96-well plates, and then the resultant cells were used for identification with GPMsDB-tk. In total, 103 colonies were successfully processed, and MALDI-TOF MS measurements by the FA heating method resulted in correct taxonomic identification for most cultures (94 correct identifications out of 103 colonies at least at the genus level), as validated by full-length 16S rRNA gene sequencing (Fig. 4, Additional file 7: Table S6). Notably, some of

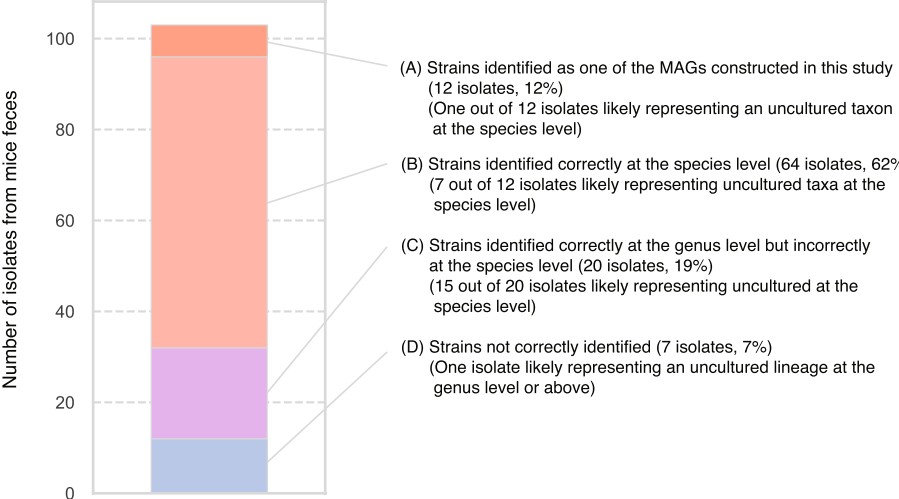

**Fig. 4** Identification of the 103 isolates obtained from mouse fecal samples based on MALDI-TOF MS profiles with GPMsDB-tk. Details of the identification results are shown in Table S6. The bar chart shows the fractions of the correct identification (categories A–C) at the genus to strain levels and incorrect identification (category D) at taxonomic levels lower than the family level. Successful genus to strain-level identifications were possible for 96 isolates, out of 103 cultures. In the fractions of categories A–C, the number of uncultured taxa at the species level were suggested based on GPMsDB identification and 16S rRNA gene sequencing. For the category (D), we identified one isolate (isolate 35), for which genomes were not available in the GPMsDB as they represent a new lineage at the genus level or above (see Table S6)

the cultured colonies ("isolates") were assigned to MALDI-TOF MS profiles from reconstructed genomes (S1_bin13_metabat, S1_bin15_metabat, and S12_bin60_concoct in Additional file 7: Table S6, matching with 12 isolates as shown in Fig. 4). This shows the benefits of adding MAGs for a sample of interest to guide cultivation efforts. Overall, 76 isolates (72%), including those matched with MAGs, were correctly identified at the species level; i.e., in these cases, MALDI-TOF MS identification with GPMsDB-tk gave the same GTDB taxonomy string down to the species level with those of 16S rRNA gene sequencing. Twenty strains out of 103 isolates (20%) were not correctly identified at the species level, but identified correctly at the genus level. Remaining 7 isolates (7%) were not correctly identified at the genus level and below; this may be due to weak MALDI-TOF MS signals due to low cell biomass and/or lack of appropriate genome sequences in the database.

## Discussion

Currently, microbial identification by MALDI-TOF MS is limited to well-characterized species, most of which are pathogens for clinical diagnostic purposes, due to a lack of comprehensive mass spectral reference libraries [13, 16]. Our toolkit (GPMsDB-tk) and database (GPMsDB) thus substantially expand its utility for not only clinical discipline but also the screening of microbes in other microbial fields, including a search for uncultured lineages of descent in the domains *Bacteria* and *Archaea*. All the MALDI-TOF MS platforms available in the market can provide "peak list" in text format from MALDI-TOF MS measurements and the toolkit can recognize such format for identification; this

should make the toolkit universally applicable to existing virtually all MALDI-TOF MS instruments.

We demonstrated the utility of our toolkit for screening uncultured strains; firstly, uncultured genomes are recovered as MAGs from metagenomes of samples of interest, and then cultivation of cells is performed in a high-throughput format, such as using 96-well plates, and resulting grown cells are subjected to MALDI-TOF MS identification with a customized protein mass database containing the MAGs. We found that MALDI-TOF MS measurements with GPMsDB-tk successfully identified the isolates in most of the determinations (96 correct calls out of 103 isolates at the genus to strain [MAG] levels, 93%), as validated by full-length 16S rRNA gene sequencing (Additional file 7: Table S6). Still, we found discrepancies between MALDI-TOF MS identification and 16S rRNA gene sequence calls at the species level (20 out of the 94 correct genus-level identifications). We thought the species-level discrepancies might occur due to the following multiple reasons; (1) lacking appropriate genomes representing species of interest in the mass database (i.e., the right species for the strain of interest), (2) lacking appropriate 16S rRNA gene sequence representing the strain in the 16S database used, and (3) the quality of the measured MALDI spectra is not high enough for species- to strain-level identification. As noted in Additional file 7: Table S6, we performed genome sequencing of isolates 35, 41, and 53, which gave incorrect identification at the species or genus level, confirming that the addition of such genomes in the mass database allowed correct identification with the corresponding genomes. Because such misidentification can be seen for strains exhibiting moderate 16S hit with GTDB (release 95) SSU rRNA gene sequences (93.4–99.7%), we thought that the majority of the misidentification is due to reasons (1) and (2). These imply the coverage of the mass database (i.e., genome sequences) is not perfect at this moment and there is a need for a continuous expansion of the database for covering entire bacterial and archaeal lineages.

Construction of similar databases for eukaryotic microbes should be possible to expand the toolkit for *Eukarya*. As we gain a deeper understanding of gene structure, gene expression, and translation/post-translation events in cells, the prediction of protein mass profiles will be further improved, and the method will become more accurate and robust.

## Conclusions

We have demonstrated the ability to accurately identify bacterial and archaeal strains at the species to subspecies levels based on a comprehensive in silico protein mass database, in which correct identification for > 90% of measured spectra was achieved and thus streamlining microbial identification for any existing bacterial and archaeal lineages. We anticipate that the toolkit described here will facilitate the screening of thousands of axenic cultures potentially containing yet-to-be cultured and/or biotechnologically relevant bacterial and archaeal cells.

## Methods

### Microorganisms

The bacterial and archaeal strains used in this study are listed in Additional file 2: Table S1; most strains were obtained from the Japan Culture of Microorganisms (JCM),

Biological Resource Center, NITE (NBRC), or Deutsche Sammlung von Mikroorganismen und Zellkulturen GmbH (DSMZ). Some strains were originally isolated and maintained in our laboratory. In total, 94 strains (84 bacterial and 10 archaeal strains) were used, representing a wide range of phylogenetically distinct bacterial and archaeal lineages (15 phyla in total). Anaerobic strains were cultivated using a basal medium described previously [17]. Anaerobic cultivations were performed at either 37 or 55℃ in 50-ml serum vials containing 20 ml of medium ($pH_{25°C}$ 7.0) under an atmosphere of $N_2/CO_2$ (80:20, v/v). Neutralized substrates (and/or hydrogen) were added to the vials prior to inoculation (Additional file 2: Table S1). Aerobic strains were grown with either 802 or 909 medium (Microlunatus Medium) described in the NBRC online catalog (https://www.nite.go.jp/nbrc/catalogue/). Growth temperature, atmosphere, basal medium, and substrate used are shown in Additional file 2: Table S1.

For some *Actinomycetota*, four different cultivation conditions were evaluated; (1) liquid-medium culture with a complex (carbohydrate-containing) medium (227 medium without agar or 802 medium without agar), (2) solid-medium culture with a complex medium (227 medium or 802 medium), (3) liquid-medium culture with a protein-based medium (230 medium without agar or DMS92 medium without agar), and (4) solid-medium culture with a protein-based medium (230 medium or DSM92 medium) (Additional file 5: Table S4). For MALDI-TOF MS analysis, grown cells were harvested at the stationary growth phase and stored in 100% ethanol at −20℃. Strains lacking public genome sequences were subjected to genome sequencing as described below and cell pellets of these strains were also stored in RNAlater Stabilization Solution (Invitrogen) at −20℃.

### Mouse fecal samples

Feces of healthy wild-type C57BL/6 mice were used as the inoculum for the cultivation of anaerobes. Fecal pellets (2–3 pellets) from two males were collected and cut into two pieces; one piece was transferred to a 1.5-ml tube with RNAlater Stabilization Solution for DNA extraction and the other piece was homogenized in another 1.5-ml tube containing the anaerobic basal medium described above, followed immediately by placing the homogenates in the medium into an anaerobic 50-ml serum vial with the same basal medium for cultivation. Feces in RNA later were stored at −20℃ until DNA extraction. All the fecal samples were mixed and used for further analyses.

### Cultivation of microbes from fecal samples

The fecal sample in an anaerobic 50-ml serum vial was immediately subjected to cultivation of bacterial cells using solid EG medium (JCM 14 medium) (http://jcm.brc.riken.jp). Incubation was performed under anaerobic conditions ($N_2/CO_2/H_2$, 92:5:3, v/v) at 37℃ for 3 days. More than 100 colonies on the plates were randomly selected and anaerobically subcultured with liquid YCFA medium (JCM 1130 medium) under the conditions described above. We note that no further purification step of colonies was performed since we sought to examine the performance of GPMsDB identification for rapid screening of colonies. The grown cells were then subjected to MALDI-TOF MS analysis and full-length 16S rRNA gene sequencing (described below).

### DNA extraction and Illumina sequencing

DNA extraction and purification from pure cultures was performed using the MagAttract HMW DNA Kit (Qiagen). Short-read sequencing libraries were prepared using the Illumina DNA Prep Kit (formerly, Nextera DNA Flex, Illumina) starting from 10 ng of input DNA and sequenced on a NextSeq 500 instrument. DNA extraction from the mixed fecal sample was performed using the method described previously [18]. Two DNA extracts were prepared from the same sample to facilitate differential coverage population genome binning [19, 20]. Briefly, the sample was divided into two fractions; one was used for DNA extraction with ISOSPIN-based bead-beating (three rounds of beating for 1 min beating each time [18]), and the remainder was used for DNA extraction with the same ISOSPIN-based method without bead-beating (the three rounds of beating were replaced with weak vortexing). The DNA concentration of the extracts was measured with a Qubit dsDNA BR Assay Kit (Thermo Fisher Scientific). DNA library preparation was performed using the SMARTEer ThruPLEX Seq Kit (TaKaRa) with Covaris-based DNA fragmentation as described previously [18], targeting an average fragment size of 300 bp. DNA sequencing of the two libraries was performed with a single sequencing run using the NextSeq 500/550 Mid Output Kit v2.5 (300 cycles). Basic statistics of all the sequencing data are shown in Additional file 3: Table S2.

### Nanopore sequencing for pure cultures

For *Geobacter hydrogenophilus* H-2, *Pseudothermotoga lettingae* TMO, and *Thermodesulfovibrio islandicus* TSL-P1, Oxford Nanopore Technologies (ONT) sequencing libraries were prepared using the SQK-PBK004 PCR Barcoding Kit. For *Agromyces rhizosphaerae* 14, *Brachybacterium conglomeratum* 5–2, *Desulforhabdus amnigena* ASRB1, *Glycomyces algeriensis* LLR-39Z-86 and *Tepidanaerobacter syntrophicus* OL, ONT sequencing libraries were prepared using the SQK-LSK109 Ligation Sequencing Kit and EXP-NBD104 Native Barcoding Expansion pack. Sequencing was performed on an R9.4.1 flow cell using an ONT MinION device. Basic statistics of the sequencing data are shown in Additional file 3: Table S2.

### Genome and metagenome assembly and binning

For the Illumina sequencing data, binary base call sequence files were converted to FASTQ format using Illumina's bcl2fastq Conversion Software (version 2.20.0.422). For the ONT sequencing data, basecalling was performed with Guppy (version 4.5.3; https://community.nanoporetech.com) using the high-accuracy model (command line flags –config dna_r9.4.1_450bps_hac.cfg), and library demultiplexing and trimming of barcodes (command line flag –trim_barcodes); short and low-quality reads (< 1000 bp and average quality of < 9) were discarded. For the Illumina sequencing data, reads were quality controlled with fastp (version 0.20.0 [21]), specifying command line flags –trim_tail1 1 –trim_tail2 1 –cut_right –cut_right_window_size 4 –cut_right_mean_quality 18 –trim_poly_x –poly_x_min_len 10 –n_base_limit 0 –low_complexity_filter –length_required 50. Long-read assembly was performed using Flye (version 2.9 [22]), with default parameters. The long-read assembly was then polished using the quality-controlled ONT reads with Racon (version 1.5.0 [23]), specifying command line flags -m 8 -x -6 -g -8 -w 500, followed by Medaka (version

1.2.3; https://github.com/nanoporetech/medaka), specifying command line flag -m r941_min_high_g360. A final polishing step was performed using the quality-controlled Illumina short reads with Polypolish (version 0.5.0 [24]), with default parameters. The strains lacked nanopore long-read data due to difficulty in cultivating them in a large-scale culture (Additional file 3: Table S2). Illumina read pairs were merged using Pear v0.9.11 [25] and quality control of unmerged pairs was performed with Trimmomatic v0.39 [26], with specifying options "LEADING:3 TRAILING:3 SLID-INGWINDOW:4:15 MINLEN:36." The resulting merged/unmerged reads were used for assembly using SPAdes v3.13.1 [27] with the options "careful" and "-k 77,99,127." For metagenomes of mouse feces, in total, 60 million read pairs and 54 million read pairs for DNA extracts with and without bead-beating were obtained and quality control of these reads was performed using fastp v0.20.0 [21] as described previously. MetaSPAdes v3.15.4 [28] was used for coassembly and assembly for individual samples with options "–only-assembler" and "-k 77,97,127". Assembly statistics are reported for contigs $\geq$ 1000 bp (Additional file 3: Table S2). Reads were mapped to contigs with Bowtie v2.4.1 [29] with default parameters and the coverage of contigs was calculated with Samtools v1.15.1 [30]. MAGs were independently recovered from assemblies for each sample using MetaBAT v2.15 [31] with the option "–minContig 1500". In parallel, Maxbin v2.2.7 [32] was used with coassembly contigs using differential coverage information for the two samples (with an option "-min_contig_length 1500"). Similarly, genome binning with Concoct v1.1.0 [33] was performed with coassembly contigs using differential coverage information for the two samples (with the option "–length_threshold 1500"). The resulting genome bins were processed with DAS tool v1.1.3 [34] with default settings to integrate the bins into an optimized, nonredundant set of bins. The completeness and contamination of the genome bins in the nonredundant set were estimated using CheckM using lineage-specific marker genes and default parameters. CoverM v0.6.1 (https://github.com/wwood/CoverM) was used to estimate the abundance (%) of each genome bin in the sample (shotgun data with bead-beating DNA extraction) with the "genome" function with Minimap2 v2.17-r941 [35] and Samtools v1.13. Kraken2 v2.1.1 [36] was used to estimate the taxonomic composition of the sample (bead-beating DNA data) with default parameters except for the option "–confidence 0.05," using the reference genome data GTDB-r202 (R06-RS202) [6]. Basic statistics of the assembly are shown in Additional file 3: Table S2.

### Full-length 16S rRNA gene sequencing

Near full-length 16S rRNA gene amplicon libraries for long-read sequencing (Oxford Nanopore Technologies) were prepared by two-step tailed PCR directly from colonies of the isolates. To this end, a loopful of cells was collected and resuspended in 20 µl of nuclease-free water. In the first round of PCRs, the V1 to V9 regions of the 16S rRNA gene were amplified using the primers 5′-TTTCTGTTGGTGCTGATATTGC<u>AGAGTTTGATCMTGG CTCAG</u>-3′ (27F forward primer, the locus-specific region is underlined) and 5′-ACTTGC CTGTCGCTCTATCTTC<u>TACGGYTACCTTGTTACGACTT</u>-3′ (1492R forward primer). Reactions (20 µl) contained 1 × Platinum SuperFi II Green PCR Master Mix, 500 nM each of the forward and reverse primers, and 1 µl of cell suspension. The thermal cycling

conditions were as follows: 98°C for 3 min; 30 cycles of 98°C for 10 s, 60°C for 10 s and 72°C for 45 s; 72°C for 5 min. Following verification of amplicon sizes by agarose gel electrophoresis, PCR products were purified using the Agencourt AMPure XP PCR Purification System (0.6 × volume of beads) and eluted in 40 µl of nuclease-free water. The second round of PCRs (25 µl) contained 1 × Platinum SuperFi II Green PCR Master Mix, 0.5 µl of barcoded PCR primers ("PCR Barcode" from ONT's PCR Barcoding Expansion 1–96, EXP-PBC096), and 1 µl of purified first-round PCR product. Amplification was performed for 8 cycles using the temperature profile described above, PCR products were purified as described above and eluted in 35 µl of nuclease-free water. Amplicon concentrations were determined using the D5000 DNA ScreenTape Assay and Agilent 4200 TapeStation, and sequencing libraries combined in equimolar proportions. The pooled libraries were then further processed using ONT's Ligation Sequencing Kit (SQK-LSK110) following the manufacturer's instructions. Sequencing was performed on an R9.4.1 flow cell using a MinION device. Two 3-h sequencing runs were performed on a single flow cell and the flow cell washed using ONT's Flow Cell Wash Kit (EXP-WSH004) between sequencing runs. Guppy (version 6.0.1; https://community.nanoporetech.com) was used for basecalling with the superaccuracy model (command line flags –config dna_r9.4.1_450bps_sup.cfg) and library demultiplexing (command line flags –require_barcodes_both_ends –trim_barcodes –barcode_kits "EXP-PBC096"). Sequences with both forward and reverse PCR primers, allowing up to 3 mismatches, anchored at the end of the reads were then identified and trimmed using Cutadapt (version 3.5 [37]); reads without identifiable primers were discarded. Reads with a length of 1300 to 1700 bp and an average quality of 12 were retained using SeqKit (version 2.2.0 [38]) and (re)oriented using VSEARCH's (version 2.18.0; [39]) orient function using the Ribosomal Database Project Training Set 18 [40]. Next, sequences from each library were individually clustered using isONclust (version 0.0.6.1 [41]), with default parameters. For each cluster, 250 sequences with the highest average base quality and a length within one standard deviation of the mean were retained. These sequences were then compared against each other using VSEARCH's allpairs_global function, specifying –id 0.9, and the sequence with the highest average identity to all other sequences was retained. This sequence was then polished with ONT's Medaka (version 1.6.1; https://github.com/nanoporetech/medaka), specifying command line flag -m r941_min_sup_g507. Polished consensus sequences were compared against the 16S rRNA gene sequences associated with GTDB r95 (R05-RS95) using VSEARCH, specifying command line flags –id 0.9 –maxaccepts 100 –maxrejects 25 –query_cov 0.95, and default identity definition. Taxonomy was assigned based on the match with the highest sequence identity, including multiple matches.

### Sample pretreatment for MALDI-TOF MS

For the preparation of MALDI-TOF MS measurement, the following four preparation methods were used: (1) simple formic acid-based method (FA), (2) bead-beating in 2,2,2-trifluoroacetic acid (TFA-beating), (3) heating in formic acid with beating in ethanol (FA heating), and (4) simple formic acid heating method (simple FA heating), which were developed based on the methods previously described [42–44]. For the first method (FA), briefly, 1–20 ml of grown culture was transferred into 1.5–15 ml tubes and centrifuged at 5000–10,000 × g for 5–10 min. Then, the supernatant was discarded, and 100–500 µL of 100% ethanol was placed on the cell pellet. After centrifugation at

10,000 $\times$ $g$ for 5 min, ethanol was discarded, and cell pellets were dried at room temperature. Ten microliters of formic acid solution (70%) was then added to the cell pellet and mixed, followed by the addition of 10 µL of acetonitrile. Vortexing was then performed for 30 min at room temperature. After centrifugation at 10,000 $\times$ $g$ for 5 min, 1 µL of the supernatant was spotted onto MALDI plates and air-dried. One microliter of alpha-cyano-4-hydroxycinnamic acid (CHCA) solution (10 mg/mL, in the presence of 50% acetonitrile and 1% trifluoroacetic acid) was used as the matrix solution. For performing TFA-beating, cell pellets were prepared, and 100% ethanol was added as described above. Then, 0.5 g of 0.1-mm autoclaved zirconia beads was added, followed by bead-beating using the FastPrep-24 instrument for 60 s at a speed of 6 m/s 3 times. After centrifugation at 10,000 $\times$ $g$ for 5 min, the ethanol was discarded, and the cell pellets were dried at room temperature. Then, 75 µL of 50% acetonitrile in 1% trifluoroacetic acid solution was added, and the mixture was homogenized. Then, the solution was transferred to a 2.0 mL screwcap tube containing 50 mg of 0.1-mm autoclaved zirconia beads, followed by bead-beating using the FastPrep-24 instrument for 60 s at a speed of 6 m/s 1−4 times. After centrifugation at 10,000 $\times$ $g$ for 5 min, 1 µL of the supernatant was spotted onto MALDI plates and air-dried. One microliter of the same matrix solution was added to the spots and air-dried. For FA heating, cell pellets were prepared, and 100% ethanol was added as described above. Then, 0.5 g of 0.1-mm autoclaved zirconia beads was added, followed by bead-beating using the FastPrep-24 instrument for 60 s at a speed of 6 m/s 3 times. After centrifugation at 10,000 $\times$ $g$ for 5 min, the ethanol was discarded, and the cell pellets were dried at room temperature. Ten microliters of 70% formic acid was added, and the mixture was homogenized. Then, the solution was heated at 50 ℃ for 5 min. Ten microliters of 100% acetonitrile was added and mixed. After centrifugation at 10,000 $\times$ $g$ for 2 min, 1 µL of the supernatant was spotted onto MALDI plates and air-dried. One microliter of the same matrix solution was added to the spots and again air-dried. For performing simple FA heating, cell pellets were prepared, and 100% ethanol was added as described above. After centrifugation at 10,000 $\times$ $g$ for 5 min, the ethanol was discarded, and the cell pellets were dried at room temperature. Ten microliters of 70% formic acid was added, and the mixture was homogenized. Then, the mixture was heated at 50 ℃ for 5 min. Ten microliters of 100% acetonitrile was then added and mixed. After centrifugation at 10,000 $\times$ $g$ for 2 min, 1 µL of the supernatant was spotted onto MALDI plates and air-dried. One microliter of the same matrix solution was added to the spots and air-dried.

### MALDI-TOF MS measurement

MALDI-TOF MS analysis was performed using an AXIMA Performance (Shimadzu/Kratos) mass spectrometer equipped with a pulsed nitrogen UV laser (337 nm) in the positive ion linear mode. Unless specified, at least four mass spectra were acquired for each sample from independent sample spots in the range of *m/z* 2000−20,000. External mass calibration was carried out using *Escherichia coli* K12 (CCUG 58987) cell homogenates prepared based on the TFA-beating method. Protein mass peaks for 47 ribosomal proteins, ranging from *m/z* 4365.34 [M + H]$^+$ to *m/z* 19888.91 [M + H]$^+$, were used with the calibration function in Shimadzu Biotech Launchpad software (v2.8). For a single MALDI spot, 121 profiles (for each, 5 shots were accumulated) were obtained at

different locations within the spot and averaged to generate an output profile. A laser power of 100–110 was used with the option "Pulsed Extraction Optimized at (DA)" of 15,000.0. After measurement, the peak list, which contained a list of detected mass peaks (*m/z*) and signal intensities, was exported for each MALDI spot using the "Export/ ASCII" function in the software.

### External MALDI-TOF MS spectrum data

The MALDI-TOF MS peak lists for *Cutibacterium acnes* strains (118 peak lists for 24 different strains) were obtained in a previous study using an AXIMA Performance (Shimadzu/Kratos) mass spectrometer [45]. Peak lists of *Acinetobacter* strains (616 peak lists for 74 different strains, SAC001) were provided by NBRC, NITE, Japan.

### Database construction

Publicly available bacterial and archaeal genomes, including single-cell genomes (SAGs) and MAGs, were downloaded from NCBI's FTP site at the time of RefSeq/GenBank 95 release [14]. The same quality control criteria used for GTDB construction [2] were applied to exclude some potentially low-quality genomes from the reference genome dataset. CheckM v1.1.2 [46] was used to estimate genome quality and assembly statistics. For all the genomes, gene identification was performed with Prodigal v2.6.3 [47] with automatic estimation of the best translation table for each genome using biolib v0.1.6 (https://github.com/dparks1134/biolib); translated amino acid sequences for all predicted protein-coding genes in a genome were used for further study. Genomes that were suggested to contain an exceptionally high number of genes per unit of genome size were excluded as an additional quality control, where genomes with a number of predicted genes per coding base in a genome > 0.0018 were flagged. Flagged genomes, except those acting as representative genomes at the species level in the GTDB r95, were removed, resulting in 193,197 genomes (190,160 bacterial and 3037 archaeal genomes) being included in the mass protein database.

For all the retained genomes, associated GTDB and NCBI taxonomies were obtained from metadata files of GTDB r95. Predicted protein sequences for all the genomes were annotated by (i) comparison against UniProKB release 2019_09 [48] using DIA-MOND v0.8.36 [49] with default settings, and (ii) annotation of proteins was also performed based on Pfam 27.0 [50] using pfam_search (https://github.com/Ecogenomics/pfam_search) with HMMER v3.1b2 [51]. Ribosomal proteins were identified based on UniProKB annotation strings and the set of Pfam models (Additional file 1: Supplementary Material 1). Resulting mature protein sequences during translocation across the cell membrane were inferred using SignalP v5.0 [52], with the command line flags "-org arch" for archaeal genomes or with both options "-org gram-" and "-org gram+" for bacterial genomes. Original and mature protein sequences were merged. "Methionine loss" was considered if the first amino acid at the N-terminus was "M" and the second was "G", "A", "S", "P", "V", "T" or "C" [53]. The mass of the resulting proteins was estimated based on the average $[M+H]^+$. The list of mass $[M+H]^+$ values for all the predicted proteins from each genome was generated, only the proteins in the range of 2000 to 15,000 Da (equivalent to their expected mass-to-charge ratio, *m/z*, in MALDI-TOF MS measurements) were retained and stored as "theoretical protein mass lists." The list

of mass values for potential ribosomal proteins for each genome was also generated and stored separately. The prediction procedure for mass values was written in part of the GPMsDB-tk (GPMsDB-dbtk) scripts developed in this study.

### Simulated protein mass peaks

Simulated protein mass lists were generated using Python's "random" function from the theoretical protein mass lists (in the range of *m/z* 2000–15,000) for *Escherichia coli* (GCF_003697165.2), *Acinetobacter calcoaceticus* (GCF_000368965.1), *Bacillus subtilis* (GCF_000009045.1), *Cutibacterium acnes* (GCF_003030305.1), and *Methanothermobacter thermautotrophicus* (GCF_000008645.1). For each genome, mass values were randomly selected to obtain 50, 100, and 200 peaks per list. In addition, "noisy" mass values were generated using the "random.uniform(2000,15000)" function in python; the random noisy values were mixed with selected theoretical mass values at 0, 20, 40, 60, 80% of the noisy values in the total number of mass values, referred to as "noise (%)". For each condition, 100 random lists were generated, and peak matching was evaluated at a mass tolerance of 200 ppm.

### Data analysis

Matching of protein mass peaks was performed using GPMsDB-tk (v1.0.1, including GPMsDB-dbtk v1.0.1 for building custom databases for user-provided genomes), developed as part of this study. GPMsDB-tk is a software toolkit for assigning taxonomic identification to user-provided MALDI-TOF mass spectrometry profiles. Details of the functionality available in the GPMsDB-tk are shown in Additional file 1: Fig. S7. GPMsDB-tk is a Python package that can be run on a regular laptop using ~5 Gb of RAM for a single thread. Peak lists (obtained in this study or from other studies) were imported into the toolkit. Peak matching was performed at a mass tolerance of 200 ppm, unless otherwise specified.

To assign of peak lists to their best match in the database, we defined a peak matching (PM) score based on the number of matched peaks between a queried MALDI-TOF MS peak list and theoretical peak list, as follows:

Scoring scheme I:

$$PM_I = \frac{\#(peaks \in Q \bigcap R)}{\#(peaks \in R)}$$

where the numerator represents the number of matched mass peaks between the measured query peak list and theoretical reference peak list at a given error tolerance for peak matching, and the denominator represents the number of mass peaks in the theoretical reference peak list. Note that the PM score cannot exceed 1 because mass peaks in the query matching multiple peaks in the reference, and vice versa, are counted only once.

Scoring scheme II:

$$PM_{II} = PM_I$$

$$if \ \#(peaks \in R) \leq 800$$

$$PM_{II} = \frac{\#(peaks \in Q \bigcap R)}{800}$$

*if* $\#(peaks \in R) > 800$

where the denominator is set to 800 (that is, the average number of mass peaks per genome in the database) for reference genomes/peak lists with more than 800 masses.

Scoring scheme III:

$$PM_{III} = \frac{\#(ribosomal\ mass\ peaks\ \in Q \bigcap R)\ \times\ w\ +\ \#(nonribosomal\ mass\ peaks\ \in Q \bigcap R)}{\#(peaks\ \in R)}$$

*if* $\#(peaks \in R) \leq 800$

$$PM_{III} = \frac{\#(ribosomal\ mass\ peaks\ \in Q \bigcap R)\ \times\ w\ +\ \#(nonribosomal\ mass\ peaks\ \in Q \bigcap R)}{800}$$

*if* $\#(peaks \in R) > 800$

where $w$ represents the weighting factor for matched mass peaks from ribosomal proteins (set to a default value of 7).

As default, the toolkit uses the scoring theme III; this scoring is used for the identification of the cultured strains, including cultures from the fecal samples. Unless specified, for taxonomic identification based on MALDI-TOF MS measurements, self-mass calibration in the GPMsDB-tk (option -aa) and -m option (-m 0) were used with the option "reps" (reference data with representative genome entries at the species level), "all" (reference data with all genome entries), or "custom" (reference data with all genome entries plus user-provided custom genome data) with the option "peak_bwf" with schoring theme III ("–score_type weighted"). GPMsDB-tk provides not only score values for hit genomes but also probability value for each score. For the calculation of the probability value, 100 non-hit genomes, where 500 or 5000 top hit genomes were excluded for selection for "reps" and "all" genomes, respectively, are randomly selected, and the score is calculated for each genome selected. After collecting the scores from the 100 genomes, the distribution of scores for the given peak list is inferred using "stats.norm.sf" function of python and the frequency of appearance of the given score is calculated as the probability value. For comparative genome analyses, average nucleotide identity (ANI) was calculated with FastANI v1.33 [54] with default settings. Mash v2.3 [15] was used to calculate mash distances with all combinations of the retained 193,197 genomes. Data visualization was performed using seaborn/matplotlib in Python. Phylogenetic trees were generated primarily using GTDBtk v1.4.1 [55] with the r95 database. The GTDBtk "align" function was used with the option "–skip_trimming" to export the amino acid sequence alignment, and the sequences were imported into an ARB database [56]. Then, the sequences of selected operational taxonomic units (OTUs) were exported from ARB with custom masks to eliminate noninformative sites of sequences for phylogenetic inference. FastTree v2.1.10 [57] was used for inferring phylogeny and trees generated in ARB.

## Supplementary Information

**Additional file 1: Supplementary Material 1.** The set of Pfam models used for extracting coding proteins likely relevant to ribosomal proteins. **Figure S1.** Relationship between average nucleotide identity (ANI) and genome-wide peak matching (PM) score. **Figure S2.** Relationship between taxonomic distance and genome-wide peak matching (PM) score. **Figure S3.** Relationship between taxonomic distance and genome-wide peak matching (PM) score for three genome assemblies with among the highest number of genes, or predicted mass peaks, in the GPMsDB (*Kosakonia* sp002886105: 8,428 predicted mass peaks with *m/z* values of 2,000–20,000; *Streptomyces violaceus*: 8,375; *Actinoplanes liguriensi*: 8,171). **Figure S4.** Effect of scoring scheme on identification accuracy as evaluated with simulated mass peak lists. **Figure S5.** Accuracy of identification using scoring scheme I for simulated peak lists with varying degrees of "noise". **Figure S6.** Accuracy of identification using scoring scheme III for simulated peak lists containing varying numbers of ribosomal protein peaks. **Figure S7.** Schematic of the GPMsDB-tk. **Figure S8.** Identification results for MALDI-TOF MS peak lists of 71 bacterial reference strains. **Figure S9.** Identification results for MALDI-TOF MS peak lists of 10 archaeal strains. **Figure S10.** Identification results for MALDI-TOF MS peak lists of 13 strains of *Actinomycetota*. **Figure S11.** Identification results for MALDI-TOF MS peak lists for 74 different *Acinetobacter* strains. **Figure S12.** Same as for Figure S11, except that all genomes in the GPMsDB were searched by specifying command line option -a as "all". **Figure S13.** Identification results for MALDI-TOF MS peak lists for 24 different strains of *Cutibacterium acnes*.

**Additional file 2: Table S1.** List of strains used in this study.

**Additional file 3: Table S2.** Assembly statistics for genomes and metagenomes.

**Additional file 4: Table S3.** GPMsDB-tk results for identification of 81 bacterial and archaeal reference strains.

**Additional file 5: Table S4.** GPMsDB-tk results for identification of 13 acidobacterial strains.

**Additional file 6: Table S5.** MAGs obtained from mouse fecal samples.

**Additional file 7: Table S6.** Identification of new isolates from the same mice fecal samples.

**Additional file 8.** Review history.

### Acknowledgements
The authors thank Mamiko Noguchi Kabasawa at AIST for her help with cultivation of aerobes and MALDI-TOF MS measurements. We also thank Kazutoshi Murotomi for his help with mouse fecal sample preparation. We thank Hiroko Kawasaki at the National Institute of Technology and Evaluation (NITE), Japan, for providing peak lists for the *Acinetobacter* strains.

### Review history
The review history is available as Additional file 8.

### Peer review information

### Authors' contributions
Conception and design: Y.S., K.T., K.T. Funding support and acquisition: K.T., S.I., Y.S. Preparation of the manuscript: Y.S., D.T. Database and toolkit design and construction: Y.S., D.T. Analyses and interpretation of data: Y.S., K.T., D.T., D.M., Y.Y. A.O. and Y.S. conducted cultivation of reference strains, MALDI-TOF MS measurements, isolation of cells from fecal samples, and 16S rRNA gene sequencing. M.H. prepared samples for DNA sequencing. D.T. assembled and analyzed genomes and 16S rRNA gene sequences. D.M. set MALDI-TOF MS and sample preparation platforms. K.T. provided **C. acnes** peak lists from a previous study.

### Funding
This study was supported in part by the program "The next-generation drug discovery and development technology on regulating intestinal microbiome" (NeDDTrim) from Japan Agency for Medical Research and Development, AMED under the Grant Number JP ae0121035h0002.

### Availability of data and materials
The genomes of the reference strains sequenced in this study have been deposited under BioProject PRJDB14918-PRJDB14932. Individual genomes have been deposited in DDBJ/ENA/GenBank, and the accession numbers are provided in Additional file 3: Table S2. The initial versions of these genomes are described in this paper. Metagenome-assembled genomes from the mouse feces and 16S rRNA gene sequences of the fecal isolates are available at Zenodo under https://doi.org/10.5281/zenodo.7707336 [58]. MALDI-MS peak lists obtained in this study are also available at Zenodo under the same DOI.
GPMsDB-tk, which consists of Python scripts and documentation, is licensed under the GNU General Public License (Version 3). GPMsDB, i.e., the in silico mass database, is licensed under the Creative Commons Attribution-ShareAlike 4.0 International. The source code and documentation are available at https://github.com/ysekig/GPMsDB-tk [59] and https://github.com/ysekig/GPMsDB-dbtk [60]. The static version of the source code used in this study is also available at Zenodo under https://doi.org/10.5281/zenodo.10002228 [61]. The database is accessible at Zenodo under https://doi.org/10.5281/zenodo.8245428 [62].

## Declarations

### Ethics approval and consent to participate
Ethics approval was not needed as mouse fecal samples were obtained from an in-house breeding mouse colony in AIST.

### Competing interests
K.T., Y.Y., S. I., and K.T. are employees of Shimadzu Corporation commercializing MALDI-TOF MS instruments and related applications. Y.S., K.T., and Y.Y. have submitted patent applications related to microbial identification based on an in silico protein mass database and pretreatment of samples for microbial MALDI-TOF MS measurement; these patents do not restrict the use of the GPMsDB and relevant toolkits for research and commercial uses.

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

## 