## [**Additional file 8.** Review history. · Genome Biology]

Review History

First round of review

Reviewer 1

Were you able to assess all statistics in the manuscript, including the appropriateness of statistical tests used? No.

Were you able to directly test the methods? No.

Comments to author:

This work deals with a very important topic. The resource created has the potential to open new avenues for very interesting experimental approaches. However, the manuscript has multiple flaws that are detailed below

- Being able to generate genome-guided MALDI profiles for MAGs is highly relevant for applications. However, MAG catalogues are known to contain a substantial fraction of artefacts, and it is not sound to include MAGs in analyses meant to validate a new approach, define robust thresholds, and test the resolution of a method.

Multiple points related to the scoring method and the resolution of the approaches:

- The authors developed a convoluted score scheming calculations to in the end focus only on ribosomal proteins mostly. While this is an explorative approach, users do not eventually know which type of scoring is used in the toolkit as it is not documented.

- L91: "genome-wide PM scores correlated well with genomic relatedness". Figure 1 does not show correlations. There is an extremely wide range of peak matching scores for each of the categories shown. Hence it is fully unclear at which level of taxonomy the approach allows precise/correct separation of genomes/profiles.

- L85-100: there is no mentioning of cut-offs after matching species to the DB for a reliable identification.

- Related to above, when using the isolates, L184 reports good identification at the genus level. I believe identification at the species level must be the target and this is not reported. Table S6 displays multiple misannotation at the species level. This must be clearly described in the main results section.

- While p-values are mentioned in Suppl. Tables and can be guessed in Figures, there is no paragraph on the methods to explain how they were calculated and what are the recommendations.

Regarding the software:

While the code is on Github with an adequate license, there are numerous issues:

* (MAJOR) the installation procedure emits warnings of code deprecation, making harder for the user to know if this was successful.

* (MAJOR) access to the database has to be requested to the authors via Zenodo.

* (MAJOR) impossible to run the identification procedure of the GPMsDB-tk (fails with NameError: name 'option' is not defined), even after attempts to modify the code manually to try making it run. The database creation functions work but are useless without the identification procedure.

* (MINOR) the installation steps on the Github omit the step of cloning the repository before the installation.

* (MINOR) software dependencies, while listed, could have been made easier to tackle via conda/docker environments.

Additional, more minor points:

- The first section of results is methods and should be moved to the corresponding section

- The authors refer multiples times in the text to "lower-quality genomes" but I could not find any definition of this (thresholds used to consider genomes to be of high quality).

- Fig 1: red lines (legend) are not visible in the figure; error bars may need to be placed before the data points for better visualization; will be clearer if mass error tolerances (color code) are defined in the figure directly

- FigS1: PM scores cannot exceed one according to the authors definitions but in this figure, they are several instances above 1

- FigS7: Several typos: "costum", "randomely", "estimimate", "propability"

- Suppl. Table S6: I believe this data requires an illustration in the main body of the manuscript (see comment above on identification at the genus Vs. species level).

- L77: "posttranslational cleavage of N-terminal methionine". Do prokaryotes do this? I thought this was only eukaryotic cells. If this is prominent in prokaryotes as well, more background and adequate literature should be listed.

- L91: define PM score here not only in M&M

- L143: I think the authors did not mean "underscoring" but a more positive word such as "highlighting"

- L182: which scoring scheme is used for the identification of MAGs?

- L435: "automatic estimation of the best translation table for each genome" supposedly in the biolib software but impossible to find the evocated procedure.

Reviewer 2

Were you able to assess all statistics in the manuscript, including the appropriateness of statistical tests used? No.

Were you able to directly test the methods? No.

Comments to author:

In this paper entitled "A large-scale genomically predicted protein mass database enables rapid and broadspectrum identification of bacterial and archaeal isolates by mass spectrometry", the authors investigated the capability of their toolkit to generate and identify MALDI-TOF MS spectra from genomic and metagenomic data. This work focuses on validating the toolkit, aiming to comprehend the quality of the predicted spectra that are synthetically generated starting from DNA sequences.

I've found the methodology adopted by the authors fascinating, aiming at reducing the time and money spent on the experimental acquisition of the same number of spectra using the standard MALDI-TOF MS procedure. Furthermore, the generation of a synthetic database for a fast screening of microorganisms whose spectra are still unknown can be helpful for the whole scientific community. Here it follows my criticism of the reported manuscript.

Major comments:

Line 81. Authors declared that the GPMsDB contains about 163 million protein mass entries, translating into 845 entries per genome. How have the proteins been selected? Obviously, this number can't represent a genome's complete repertoire of proteomic sequences. Please add information about the selection of the included spectra.

Page 7. It is not clear why the authors selected microorganisms that belong to the Actinobacteria to test different sample treatments. Why Actinobacteria and not another group of microorganisms? This notion should be clarified. Furthermore, in this chapter, no data is reported. If this chapter intends to declare the best protocol, actual data should be reported and justified. Similarly, the following chapter describing the performance of the toolkit on Cutibacterium and Acinetobacter should be better written to clarify the microorganisms' choice, such as genetics, unexplored species, or whatsoever.

Page 11. The description regarding the cultivation of microorganisms is generic and not well described. For example, we don't know about incubation time and what has been done with the microorganisms' growth in EG medium. Please clarify the cultivation steps adding more information.

Regarding the availability of the toolkit, database and datasets. I had access to the toolkit via GitHub and to the evaluation datasets on Zenodo by searching "GPMsDB". However, what I could not find was the actual database of spectra. Is the R01-RS95 whit restricted access? Are the authors intentioned to release the complete database to the community? Are these profiles utilizable on a specific MALDI-TOF MS brand/software suite? Or are they "universally"

utilizable? These answers should be addressed to understand the actual benefits of the predicted spectra for the scientific community.

Minor comments:

Line 68. Several? How many?

Line 185. More data about these "incorrect identification" should be reported.

Line 187. The thermos "some" is not clear. Please report a value.

Line 202. Accurately in what extent? Please report a value.

Line 268. This sentence is a repetition of what already stated four sentence above. Please remove.

Reviewer reports:

Reviewer #1: This work deals with a very important topic. The resource created has the potential to open new avenues for very interesting experimental approaches. However, the manuscript has multiple flaws that are detailed below:

- Being able to generate genome-guided MALDI profiles for MAGs is highly relevant for applications. However, MAG catalogues are known to contain a substantial fraction of artefacts, and it is not sound to include MAGs in analyses meant to validate a new approach, define robust thresholds, and test the resolution of a method.

> Thank you for this pertinent comment. We agree that MAGs and SAGs can potentially contain artefacts that may compromise our analyses. To address this concern, we additionally tested the relationship between Mash distance (or taxonomic distance) and peak matching (PM) score using a database with only isolate genome entries (i.e., without MAG/SAG entries). For this analysis, we used “ncbi_category” in “bac120_metadata_r95.tsv” and “ar122_metadata_r95.tsv” in the GTDB metadata (<https://gtdb.ecogenomic.org/downloads>) to select for isolate genomes. In our full database, 28,600 genomes originated from environmental samples; i.e., these are considered MAGs or SAGs, out of 193,197 genome entries in total (14.8% of the total genomes in the mass database). Using only isolate genomes, we newly generated the same figures as Fig. 1 using exactly the same calculation methods/parameters. In general, overall trends (figure R1B and R1D) shown as box plots in these figures are similar to Fig. 1 in the original manuscript (redrawn as figures R1A and R1C); indicative of minor impacts of MAGs/SAGs inclusion in the original database. Please note the figures were generated with 200 randomly selected genomes for each dataset (the selected genomes are different for each dataset) and matched against all genome entries in the database (193,197 reference genome entries for original [figure R1A and R1C], 164,597 genome entries for calculation without MAG/SAGs [figure R1B and R1C]).

Figure R1, Relationship between Mash distance (A) or taxonomic distance (B) and peak matching (PM) score for the entire dataset (panels A and C) and the subset of the dataset excluding MAGs/SAGs (panels B and D). For panels A and C, the inverse Mash distance ($[1 - \text{mash}] \times 100$, nearly equivalent to ANI (%)) was calculated and binned as shown on the x-axis. For panels B and D, taxonomic distances were obtained based on the GTDB r95 taxonomy; labels on the x-axis indicate taxonomic levels at which differences in PM scores are evaluated (for example, "strain" indicates pairs of genomes within the same species). PM scores were obtained from 200 randomly selected genomes matched against all genome entries in the GPMsDB (193,197 reference genome entries for panels A and C; 164,597 reference genome entries for calculation without MAG/SAGs for panels B and D). Mass error tolerances of 50 (blue) and 200 ppm (red) were used for peak matching, and the resultant values were plotted as dots for given pairs of genomes. The distribution of the quantitative data is also shown as box plots, where the quartiles of the dataset are shown as boxes, while the whiskers extend to show the rest of the distribution, except for points that were determined to be "outliers" using a method that is a function of the interquartile range.

Now, we also stress that genomes (for isolate genomes, MAGs, and SAGs) included in our database were selected following the criteria of the developers of the widely accepted Genome Taxonomy Database (GTDB; Parks, et al., Nature Biotech. 2020) to eliminate potentially poor-quality genomes. Genome exclusion criteria were as follows: (1) completeness of <50%, (2) contamination of >10%, (3) completeness – 5 × contamination of <50, (4) >1,000 contigs, (5) N50 of <5 kbp, (6) >100,000 ambiguous bases, or (7) <40% of the 120 bacterial or 122 archaeal proteins used for phylogenetic inference. In addition, we used a criterion about coding regions as the number of genes per coding base >0.00180. These criteria were indicated in the Method section of the original manuscript. Using this approach, we demonstrated reliable identification of cultured strains, with minimal misidentification rates. Further, we used the same completeness and contamination criteria to select our own MAGs to include them in the custom database for demonstration purposes (page 9, lines 192-193 in the revised manuscript); resulting in good identification too. We thus argue that the criteria used to select genome sequences for the database construction are sufficient for our purposes, including for MAGs. Further, we believe that the inclusion of MAG/SAGs is indispensable to fully cover entire bacterial and archaeal lineages. In fact, more than 40% of species-level taxa are represented only by MAG/SAGs in the database, and to evaluate the utility of the database, such as the level of misidentification due to higher level of the coverage of the lineages in the database.

To make this point clearer we modified the sentences in the revised manuscript as "After screening for potentially lower-quality genome sequences (see "Database construction" section of the Method section in details), a total of 193,197 genomes from 190,160 bacteria and 3,037 archaea were retained (31,790 species-level taxa)." (Page 4, lines 75-77 in the revised manuscript). We also stated about the criteria in the section "Cultivation, isolation, and identification of strains from mouse faecal samples" in the Result section as "We performed shotgun metagenomics on faecal samples from mice and reconstructed genomes, which yielded 84 high-quality genome bins with genome completeness of >50% and contamination of <10% (Supplementary Table S5)" (page 9, lines 192-193 in the revised manuscript).

Multiple points related to the scoring method and the resolution of the approaches:

- The authors developed a convoluted score scheming calculations to in the end focus only on ribosomal proteins mostly. While this is an explorative approach, users do not eventually know which type of scoring is used in the toolkit as it is not documented.

- > The published version of the toolkit uses the scoring theme III as default. This has been clarified in the text as "As default, the toolkit uses the scoring theme III" (Page 25, line 573) and Supplementary Figure 7 in the revised manuscript.

- L91: "genome-wide PM scores correlated well with genomic relatedness". Figure 1 does not show correlations. There is an extremely wide range of peak matching scores for each of the categories shown. Hence it is fully unclear at which level of taxonomy the approach allows precise/correct separation of genomes/profiles.

> *Thank you for this comment. The intent of this figure was to demonstrate the relationship between PM scores and genomic relatedness (ANIs or taxonomic distances). The statement "... correlated well with ..." may here have been somewhat confusing since we did not report any correlation coefficients, etc., so we reworded that section as "genome-wide peak matching (PM) score, which is based on the number of matched peaks between mass protein peak profiles between two genomes (see more details in Method section), has a strong relationship with genomic relatedness, in terms of both average nucleotide identity (ANI, or values estimated based on Mash distances) and taxonomic distance as summarized in the GTDB" (page 5, lines 95-99 in the revised manuscript). Using all predicted proteins across genomes, the data illustrate that PM scores decrease rapidly with increasing ANI or taxonomic distance. Based on the distributions, we argue that this demonstrates that PM scores provide a reliable basis for identifying genomes belonging to the same species or genus as the query genome. As described below, this is possible because our toolkit identifies best-matching genomes for taxonomic assignment.*

- L85-100: there is no mentioning of cut-offs after matching species to the DB for a reliable identification.

> *Thank you for your comment. In response, we point out that for identification purposes, no PM thresholds are enforced and MALDI-TOF MS profiles are assigned to the genome with the highest PM score. As pointed out in the original manuscript, this approach works well if the mass database contains representative peak lists matching microorganisms to be identified at the genus-to-strain level. To estimate confidence in the accuracy of the assignments, our toolkit further compares the PM scores of a given peak-list under analysis to its best-matching genome to PM scores with 100 randomly selected non-hit genomes. Based on these data, a confidence level is then assigned as described in more detail below in response to another comment/question.*

- Related to above, when using the isolates, L184 reports good identification at the genus level. I believe identification at the species level must be the target and this is not reported. Table S6 displays multiple misannotation at the species level. This must be clearly described in the main results section.

> *Thank you for this comment. Yes, it is correct that 20 out of 103 isolate measurements (21%) showed species-level discrepancies between MALDI-TOF MS-based identification and the best match of the full-length 16S rRNA gene sequences of the isolates (as generated by nanopore sequencing). As described in the Results section, we consider the resolution of the MALDI identification to be in the range of genus to strain. We thought the species-level discrepancies might occur due to the following reasons; (1) lack of appropriate genomes representing species of interest in the mass database (i.e., the right species for the strain of interest), (2) lack of appropriate 16S rRNA gene sequence representing the strain in the 16S database used (GTDB), and (3) the quality of the measured MALDI spectra is not high enough for species- to strain-level identification.*

In order to, at least partially, resolve these uncertainties, we had performed whole genome sequences for three isolates (isolates 35, 41, and 55 in Supplementary Table S6) and showed that the addition of their predicted peak lists to the database allowed correct identification. Further, because misidentifications are mostly for strains with moderate 16S hit with GTDB (release 95) SSU rRNA gene sequences (93.4-99.7% identity), we think that the majority of the misidentification is due to reasons (1) and (2). This implies that the coverage of the mass database (i.e., genome sequences) is not perfect at this moment and there is a need for a continuous expansion of the database for covering entire lineages of prokaryotes. To make these points clearer, we added more descriptions of the

results of Supplementary Table S6 in the result section (page 9, lines 203-210 in the revised manuscript) and discussion section (pages 10-11, lines 222-243 in the revised manuscript).

- While p-values are mentioned in Suppl. Tables and can be guessed in Figures, there is no paragraph on the methods to explain how they were calculated and what are the recommendations.

> We now describe the procedure for calculating probability values in the Methods section. In short, 100 non-hit genomes are randomly selected, and the PM scores between the MALDI-TOF MS profile under analysis against these genomes are calculated. Based on the distribution (mean and sd) of the PM scores (page 24, lines 577-586), we then calculated the probability of observing the score assigned to the hits. Because of the traits of MALDI-TOF MS data, such as the contents of peak lists varies depending on the measurement conditions even for measuring the same strain, it is not easy to set recommended values for each result. However, we tried to provide our empirical “guess” as the “likelihood (%)” of a correct call in the output of the toolkit; “likelihood” (ranging <50% to 99%) is provided based on the number of ribosomal protein hits and probability value. This is also written on the Github site (<https://github.com/ysekig/GPMsDB-tk>).

Regarding the software:

While the code is on Github with an adequate license, there are numerous issues:

* (MAJOR) the installation procedure emits warnings of code deprecation, making harder for the user to know if this was successful.

> Thank you for taking the time to evaluate our software. The reported warnings do not have any impact on the installation process and functionality of the software. To prevent potential uncertainty for the users, in the new release of our toolkit (version 1.0.1 on Github), we now added the description about deprecation warning like “easy_install command is deprecated” during installation in the Github site as “During the installation, you may see some deprecation warnings like “easy_install command is deprecated” but this will not cause any issues for GPMsDB-tk” (<https://github.com/ysekig/GPMsDB-tk>).

* (MAJOR) access to the database has to be requested to the authors via Zenodo.

> Actually we didn't want to implement such restriction in Zenodo but the access control is the default in Zenodo for our license conditions. We have changed our licence to an open source license (CC BY-SA 4.0) and modified the condition accordingly. We believe that now there are no restrictions for downloading the data.

* (MAJOR) impossible to run the identification procedure of the GPMsDB-tk (fails with NameError: name 'option' is not defined), even after attempts to modify the code manually to try making it run. The database creation functions work but are useless without the identification procedure.

> Thank you very much for noticing these issues with the source code. We now carefully checked all the source, particularly main.py, common.py, defaultValues.py, and searchbest.py, and resolved such issues. The new source code is available as version 1.0.1 on Github.

* (MINOR) the installation steps on the Github omit the step of cloning the repository before the installation.

> The tool's README now describes the Github repository cloning step. “git clone <https://github.com/ysekig/GPMsDB-tk>; cd GPMsDB-tk;python setup.py install” (<https://github.com/ysekig/GPMsDB-tk>). Similarly, we modified the description for GPMsDB-dbtck too (<https://github.com/ysekig/GPMsDB-dbtck>).

* (MINOR) software dependencies, while listed, could have been made easier to tackle via conda/ docker environments.

> Thank you for your suggestions. We agree that providing Conda/Docker install options will be beneficial and we plan to release future versions as Conda/Docker packages.

Additional, more minor points:

- The first section of results is methods and should be moved to the corresponding section

> *The first section is intended to briefly explain the flow of the database construction and the resulting contents of the database. We think the section is indispensable for understanding the basic idea of the database construction and therefore would like to retain the section as original. More detailed procedures for database construction are described in the method section.*

- The authors refer multiples times in the text to "lower-quality genomes" but I could not find any definition of this (thresholds used to consider genomes to be of high quality).

> *Thank you for this comment. As described above, we used several criteria to judge the quality of genomes according to the GTDB (Parks, et al., Nature Biotech. 2020) and the number of coding regions per nucleotide. We consider genomes that did not pass these criteria as "lower-quality genomes", and these are excluded from the database. To make this point clearer, we modified the expression as "After screening for potentially lower-quality genome sequences (see "Database construction" section in Methods for details)" in page 4, line 76 in the revised manuscript.*

- Fig 1: red lines (legend) are not visible in the figure; error bars may need to be placed before the data points for better visualization; will be clearer if mass error tolerances (color code) are defined in the figure directly

> *Thank you for notifying this issue. We eliminated the red lines in the revised figures and the corresponding descriptions in the legend. To improve the visibility of the boxplots' whiskers, we reduced the symbol sizes and jitter width. We also added the definition of color code in the figure directly in the revised manuscript.*

- FigS1: PM scores cannot exceed one according to the authors definitions but in this figure, they are several instances above 1

> *Thank you very much for noticing this. In fact, we had used several scoring themes during the development of the toolkit and some of Supplementary Figures in the original manuscript were calculated using these different scoring themes. As these were not described in the manuscript and to avoid confusion, we replaced all the relevant figures generated with the scoring theme described in the legend.*

- FigS7: Several typos: "costum", "randomely", "estimimate", "propability"

> *Thank you very much for pointing out these typos. Now we carefully checked the spelling and modified Supplementary Figure S7.*

- Suppl. Table S6: I believe this data requires an illustration in the main body of the manuscript (see comment above on identification at the genus Vs. species level).

> *Thank you for your valuable comments. Now we newly generated a figure shown below for summarizing the result of Supplementary Table S6 in aspects of the correctness of the MALDI identification. Please see Figure 4 in the revised manuscript.*

Figure 4 newly generated for revision.

- L77: "posttranslational cleavage of N-terminal methionine". Do prokaryotes do this? I thought this was only eukaryotic cells. If this is prominent in prokaryotes as well, more background and adequate literature should be listed.

> *A comprehensive review is available for the posttranslational events by aminopeptidases in prokaryotic cells, Gonzales, T. & Robert-Baudouy, J. Bacterial aminopeptidases: Properties and functions. FEMS Microbiol. Rev. 18, 319–344 (1996). Several articles previously showed the cleavage in Gram-positive and negative cells, such as Rapid Commun Mass Spectrom 2006;20(24):3789-98, J Proteome Res 2007 6(10):3899-907, Anal Chem 2007 79(22):8712-9. In the revised manuscript, we cited "Wingfield, P. T. N-Terminal Methionine Processing. Curr. Protoc. Protein Sci. 88, 6.14.1-6.14.3 (2017)." In the Method section (page 22, line 518 in the revised manuscript) and modified the expression in the Result section as "accounting for posttranslational cleavage of N-terminal methionine and signal peptides (see Method for details)" (page 4, lines 78-79 in the revised manuscript).*

- L91: define PM score here not only in M&M

> *We added a brief explanation of the PM scores in the Results section according to the comment by the reviewer (Page 5, lines 95-96 in the revised manuscript).*

- L143: I think the authors did not mean "underscoring" but a more positive word such as "highlighting"

> *Thank you for your comment. We have modified the expression as the reviewer suggested (page 7, line 149 in the revised manuscript).*

- L182: which scoring scheme is used for the identification of MAGs?

> *We used the scoring theme III for all the identification of cultured strains. We have clearly described the method used for the identification of MAGs in the Method section (page 25, lines 573-574 in the revised manuscript).*

- L435: "automatic estimation of the best translation table for each genome" supposedly in the biolib software but impossible to find the evocated procedure.

> *This functionality is part of biolib/external/prodigal.py – in short, gene calling is performed with two translation tables (namely, 4 and 11) and then the translation table yielding the highest coding density selected (<https://github.com/donovan-h-parks/biolib/blob/master/biolib/external/prodigal.py>).*

Reviewer #2: In this paper entitled "A large-scale genomically predicted protein mass database enables rapid and broadspectrum identification of bacterial and archaeal isolates by mass spectrometry", the authors investigated the capability of their toolkit to generate and identify MALDI-TOF MS spectra from genomic and metagenomic data. This work focuses on validating the toolkit, aiming to comprehend the quality of the predicted spectra that are synthetically generated starting from DNA sequences. I've found the methodology adopted by the authors fascinating, aiming at reducing the time and money spent on the experimental acquisition of the same number of spectra using the standard MALDI-TOF MS procedure. Furthermore, the generation of a synthetic database for a fast screening of microorganisms whose spectra are still unknown can be helpful for the whole scientific community. Here it follows my criticism of the reported manuscript.

Major comments:

Line 81. Authors declared that the GPMsDB contains about 163 million protein mass entries, translating into 845 entries per genome. How have the proteins been selected? Obviously, this number

can't represent a genome's complete repertoire of proteomic sequences. Please add information about the selection of the included spectra.

> *Thank you for this comment. Yes, it is correct that the GPMsDB does not contain all predicted proteins. More specifically, given the size range of typical MALDI-TOF MS measurements, the GPMsDB contains predicted proteins only with theoretical molecular masses in the range of 2,000 to 15,000 Da (equivalent to their expected mass-to-charge ratio, m/z, in MALDI-TOF MS measurements) from all the predicted proteins in a single genome. This was described in the first section of the Result section (more details are written in the Method section as well) but was not detailed in the Method section. To make this point clearer, we added more detailed explanations about the selection in the Method section too (page 22, lines 504-507 in the revised manuscript).*

Page 7. It is not clear why the authors selected microorganisms that belong to the Actinobacteria to test different sample treatments. Why Actinobacteria and not another group of microorganisms? This notion should be clarified. Furthermore, in this chapter, no data is reported. If this chapter intends to declare the best protocol, actual data should be reported and justified. Similarly, the following chapter describing the performance of the toolkit on Cutibacterium and Acinetobacter should be better written to clarify the microorganisms' choice, such as genetics, unexplored species, or whatsoever.

> *Thank you for the comments. Actinobacteria, such as members of the genus Streptomyces, are typical Gram-positive bacteria that are generally considered as difficult to lyse and make this be more challenging for MALDI-TOF MS sample prep. To make this point clear, we have added sentences to explain the rationales for selecting the lineage as "13 actinobacterial strains as examples of Gram-positive-type cell wall bacteria, which are generally thought to be difficult to lyse" (page 7, lines 155-157 in the revised manuscript). As for the reported results, we showed results in Supplementary Figure S10 and Supplementary Table S4 in the original manuscript. Now, we updated Supplementary Figure S10 with a summary table / heatmap showing the rate of correct identification using the different methods. According to the reviewer's suggestion, we also added a sentence to describe why we used Cutibacterium and Acinetobacter strains as "We further assessed the performance of our toolkit by applying it to two sets of externally generated MALDI-TOF MS datasets that were obtained externally, as additional examples" (page 8, lines 176-177 in the revised manuscript). In addition, we have changed the subtitle of the section as "Identification using externally obtained MALDI-TOF MS profiles" (page 8, lines 175 in the revised manuscript).*

Page 11. The description regarding the cultivation of microorganisms is generic and not well described. For example, we don't know about incubation time and what has been done with the microorganisms' growth in EG medium. Please clarify the cultivation steps adding more information.

> *Thank you for the comments. We have added more sentences to describe the cultivation steps in the Method section (page 13, lines 297-300 in the revised manuscript).*

Regarding the availability of the toolkit, database and datasets. I had access to the toolkit via GitHub and to the evaluation datasets on Zenodo by searching "GPMsDB". However, what I could not find was the actual database of spectra. Is the R01-RS95 whit restricted access? Are the authors intentioned to release the complete database to the community? Are these profiles utilizable on a specific MALDI-TOF MS brand/software suite? Or are they "universally" utilizable? These answers should be addressed to understand the actual benefits of the predicted spectra for the scientific community.

> *We are intended to release the database for the entire scientific community as long as the license conditions apply. As indicated by avobe, we didn't want to implement to some extent a "restricted access" for the database in Zenodo but the access control is the default in Zenodo for our license conditions. We have modified the license as an open source license compliant. We believe that now there are no restrictions for downloading the data.*

To our knowledge, all MALDI-TOF MS platforms available in the market can generate “peak-list” data (in a text file) from MALDI spectrum obtained. To make this point clear, we have added a sentence to explain this point in page 10, lines 218-221 in the revised manuscript.

Minor comments:

Line 68. Several? How many?

> We have modified the expression as “the identification of 103 cultured strains” in page 3, lines 68-69 in the revised manuscript.

Line 185. More data about these "incorrect identification" should be reported.

> Thank you for the comment. Now we newly made a figure (Figure 4) to summarize the number (rates) of the correct identification at different taxonomic levels.

Line 187. The thermos "some" is not clear. Please report a value.

> Thank you for the comment. This is also clarified by providing Figure 4. Also, we have added more sentences to describe these points in the Result section (page 9, lines 203-210 in the revised manuscript).

Line 202. Accurately in what extent? Please report a value.

> According to the suggestion by the reviewer, we have added quantitative aspects of the point in the sentence as “We have demonstrated the ability to accurately identify bacterial and archaeal strains at the species to subspecies levels based on a comprehensive in silico protein mass database, in which correct identification for >90% of measured spectra was achieved and thus streamlining microbial identification for any existing bacterial and archaeal lineages” (page 11, lines 250-253 in the revised manuscript).

Line 268. This sentence is a repetition of what already stated four sentence above. Please remove.

> Thank you for the notification. We have removed parts of the relevant sentences as the reviewer suggested.

Second round of review

Reviewer 1

- Fig 1: despite the change in wording in the text (strong relationship), the data, specifically the right panel (taxonomic level) is not really convincing; there is wide variation of PMs within each taxonomic rank and a visible increase in score for the species level. Does not look like a "strong relationship" in my opinion and the authors should elaborate on this in the text. A statistical method to demonstrate significance would be a less subjective approach.
- Methods: the procedure to obtain and identify isolates must be described in more detail; how was culture purity ensured (e.g. restreaking how many times) and observed? Database and sequence similarity thresholds used to identify the isolates must be mentioned L302
- The authors may need to use the updated Phylum nomenclature, e.g. Actinomycetota, not Actinobacteria, etc.

Reviewer 2

All my comments have been addressed. The revision has been taken seriously, resulting in a substantial enhancement of the final manuscript's quality.

Authors Response

Point-by-point responses to the reviewers' comments:

Reviewer #1:

- Fig 1: despite the change in wording in the text (strong relationship), the data, specifically the right panel (taxonomic level) is not really convincing; there is wide variation of PMs within each taxonomic rank and a visible increase in score for the species level. Does not look like a "strong relationship" in my opinion and the authors should elaborate on this in the text. A statistical method to demonstrate significance would be a less subjective approach.

> *Thank you very much for this comment. As shown in Figures S1 and S2, in general, we thought that there are strong relationship between PM scores and genomic ANI (or taxonomic levels), in particular when we look at a single query genome. However, some genomes (with higher numbers of genes in their genomes, Figure S3) showed a relatively narrow range of PM score changes along with ANI and taxonomy differences. Figure 1 shows overall trends of such relationships, making the figure wide variations of PM scores within each taxonomic rank, as the reviewer notified. Now we have carefully described about these trends in the text and avoid using a strong statement based solely on Figure 1. Now we have modified the relevant sentences in the result section as follows "As shown in Fig. 1 (also see Supplementary Additional file 1: Figs. S1, S2, and S3), genome-wide peak matching (PM) score, which is based on the number of matched peaks between mass protein peak profiles between two genomes (see more details in Method section), was related with genomic relatedness, in terms of both average nucleotide identity (ANI, or values estimated based on Mash15 distances) and taxonomic distance as summarized in the GTDB2,6. As shown in Additional file 1: Figs. S1, S2, PM scores had a strong relationship*

with ANI values above 80%, as well as with taxonomic ranks at the genus to strain levels. We note that relatively wide range of PM scores seen in Fig. 1 is due to differences in the relationship between PM scores and genomic relatedness among genomes, especially for genomes with a high number of genes (Additional file 1: Fig. S2)” (page 5 in the revised ms with changes highlighted). We believe the modifications addressed the queries given by the reviewer.

- Methods: the procedure to obtain and identify isolates must be described in more detail; how was culture purity ensured (e.g. restreaking how many times) and observed? Database and sequence similarity thresholds used to identify the isolates must be mentioned L302.

> *We have added a sentence in the Method section to further describe the isolation procedure as “No further purification step was performed since we sought to examine the performance of GPMsDB identification for rapid screening of colonies” (page 14 in the revised ms with changes highlighted). For 16S rRNA gene-based identification, we have described them in the original manuscript in the “Full-length 16S rRNA gene sequencing” section of the Method section.*

- The authors may need to use the updated Phylum nomenclature, e.g. Actinomycetota, not Actinobacteria, etc.

> *Thank you for the valuable comment. Now we have updated the nomenclature of the phylum Actinomycetota throughout the text and figures.*

Reviewer #2:

-All my comments have been addressed. The revision has been taken seriously, resulting in a substantial enhancement of the final manuscript's quality.

> *Thank you again for all the valuable comments from the reviewer.*